# Towards Flexible Evaluation for Generative Visual Question Answering

## ABSTRACT

Throughout rapid development of multimodal large language models, a crucial ingredient is a fair and accurate evaluation of their multimodal comprehension abilities. Although Visual Question Answering (VQA) could serve as a developed test field, limitations of VQA evaluation, like the inflexible pattern of Exact Match, have hindered MLLMs from demonstrating their real capability and discourage rich responses. Therefore, this paper proposes the use of semantics-based evaluators for assessing unconstrained open-ended responses on VQA datasets. As characteristics of VQA have made such evaluation significantly different than the traditional Semantic Textual Similarity (STS) task, to systematically analyze the behaviour and compare the performance of various evaluators including LLM-based ones, we proposes three key properties, i.e., Alignment, Consistency and Generalization, and a corresponding dataset Assessing VQA Evaluators (AVE) to facilitate analysis. In addition, this paper proposes a Semantically Flexible VQA Evaluator (SFVE) with meticulous design based on the unique features of VQA evaluation. Experimental results verify the feasibility of model-based VQA evaluation and effectiveness of the proposed evaluator that surpasses existing semantic evaluators by a large margin. The proposed training scheme generalizes to both the BERT-like encoders and decoder-only LLM.[1]

## CCS CONCEPTS

• **Computing methodologies** → **Scene understanding**; *Activity recognition and understanding*; Information extraction.

## KEYWORDS

Visual Question Answering, Semantic Textual Similarity, Contrastive Learning, Evaluation Method

## 1 INTRODUCTION

Visual Question Answering (VQA) evaluates the multimodal comprehension abilities by posing questions about given images and comparing the model's responses with annotated answers[22, 24, 25, 38, 40, 46, 51]. However, current VQA evaluation metrics have made it tough for evaluating the rich responses of Multimodal Large Language Models (MLLMs).

[1]All related codes and data will be released.

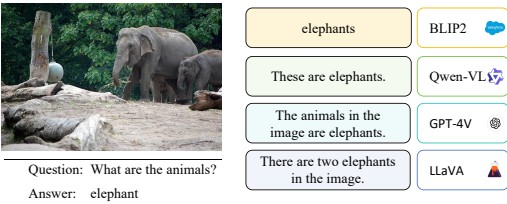

Figure 1: Responses from four MLLMs on a simple visual question. The responses are different in length, styles and complexity, which can all be considered correct but none of them exactly matches the annotated answer.

Most VQA datasets comply with a triplet format and each sample consists of a question, image and annotation. Annotations are often a single word or phrase [25, 38, 46, 51, 52] or a set of ten candidate answers [22, 24, 40, 47]. Current evaluation metrics, Exact Match [38] (for samples without candidate answers) and VQA Score [6] (for samples with ten candidate answers), both require the responses to be identical in morphology with the annotation to be considered correct. Variations in tense, singular or plural forms and synonyms are not allowed, let alone sentence-style responses from MLLMs.

Traditional vison-language models treat VQA as a classification problem [43, 46, 49, 56, 57], where answers collected from the training set are used to establish pre-defined classes, and the possible responses are constrained to these classes. Thus the problem of evaluating multifarious responses does not exist. However, MLLMs treat VQA as a generative problem [8, 18, 30, 34] and generates assorted responses. Meanwhile, the growing trend that the MLLM community prefers zero-shot test, has made it even tougher for models to generate responses that are identical to the ground-truth answers. As shown in Figure 1, semantically equivalent but morphologically distinct responses are not accepted.

Although it is possible to force the model to output a single word with a harsh prompt, such remediation may potentially damage the performance and make it unfair for different MLLMs, especially for those with poor instruction-following ability and those that tend to response with long sentences. As MLLMs inherit the in-context learning capability of LLMs, it is feasible to introduce in-context examples to force short responses. However, since different MLLMs contain different in-context learning capability, such practice interferes a fair evaluation of MLLMs' multimodal comprehension performance. There thus is an urgent need for a metric that aligns well with human judgment and accommodates various response types while ensuring consistent evaluation despite variations in response morphology.

To compare different evaluators, traditional Semantic Textual Similarity (STS) task measures the difference between predicted scores and human annotation results from a single aspect of semantic relevance. However, both intuition and our experiments (refer

to Section 3.1 and 5.3) suggest that there is a significant difference between the evaluation of correctness in VQA responses and the traditional assessment of media-text relevance in STS.

Therefore, to systematically evaluate the performance of an evaluator, with the unique characteristics in the task of VQA response evaluation taken into consideration, we propose three quantitative key properties, i.e., **Alignment**, **Consistency** and **Generalization**. Alignment stands for the overall correspondence of predicted scores with human annotation. Consistency measures how well an evaluator accommodates semantically equivalent responses of different morphology and length. Generalization indicates the variance of performance on different sources of data. Further, to facilitate comprehensive analysis of the performance and behaviour of evaluators, we provide a human-annotated dataset Assessing VQA Evaluators (AVE) that grades the correctness of model responses towards ground-truth labels on VQA datasets. AVE is further augmented by ChatGPT and WordNet [41] to increase the diversity.

As pilot experiment shows (refer to Section 5.3), formulaic metrics (BLEU [44], ROUGH [33], METEOR [10]) and model-based metrics [14, 21, 32, 45] perform poorly on the VQA response evaluation task. Therefore, we propose a novel evaluator that is trained with meticulously designed pretraining tasks. The tasks are designed for improving the embedding representation of VQA text, which utilizes contrastive learning to guide the evaluator to capture the fine-grained difference within a text pair and ignore the noise in morphology and length. Experiments demonstrate that the proposed pretraining tasks significantly improve the performance of our evaluator on the AVE dataset, making the evaluator's prediction aligns much better with human judgement.

The contribution of this paper can be concluded as follows:

- This paper addresses the dilemma, where rich responses of MLLMs hinder fair evaluation under current metric, by proposing semantic-similarity-based evaluation that applies to various VQA responses.
- This paper proposes three quantitative key properties in VQA response evaluation based on its characteristics, and a high-quality human-annotated dataset, AVE, for assessing different evaluators comprehensively. In addition, we evaluate the performance of various types of existing semantic similarity evaluators on the proposed AVE dataset.
- Experimental results demonstrate the feasibility of applying model-based methods to the flexible evaluation of VQA responses as well as the effectiveness of our proposed evaluator. Our evaluator significantly surpasses existing methods, including ChatGPT and the SOTA embedding model Voyage-lite-02-Instruct [2] by a large margin. Our training scheme generalizes to both the encoder-only and decoder-only models.

## 2 RELATED WORK

### 2.1 Visual Question Answering

As the answer space of most open-ended VQA datasets is limited and the same answer applies for multiple questions (the most common 2,000 answers in the training set of VQA v2[22] is able to cover about

94% questions in its validation set), early methods [43, 46, 49, 56, 57] treat VQA as a classification task, which adopt answers in the training set as class labels and train with classification loss. The limited answer space of such approach makes it unable to predict unseen classes and limits the generalization, which inspires the utilization of generative methods on VQA [8, 18, 30, 34, 35] and facilitates responses on a more open vocabulary.

### 2.2 Semantic Textual Similarity

Current semantic evaluation tasks include Semantic Textual Similarity (STS) [4, 5] that assesses to what extent the two sentences are related, Paraphrase Identification [48, 59] that decides whether two texts express the same meaning, and Natural Language Inference [17, 54] that determines the logical relationship between texts. The essence of these tasks lies in quantifying the degree of semantic equivalence between sentences, which is a fundamental challenge due to the complexity and variability of natural language. Methods in STS include formulaic methods like BLEU [44], ROUGH [33], METEOR [10] and model-based ones [21, 32, 42, 45, 61]. The former mainly relies on n-gram or other statistic features between the candidate and reference (which corresponds to the question-answer pair and question-response pair in this paper) to calculate the overlap and import penalty for noise. The latter utilizes models as encoders to extract the information and compare between the candidate and reference. Early model-based evaluator [61] compares the similarity of each pair each time, which is computation-consuming. Later works [21, 32, 42, 45] first generate embeddings separately for the candidate and reference, then simply calculates the cosine similarity between embeddings as the similarity score. Due to the style of STS, either formulaic or model-based methods pay more attention to the overall similarity and are less capable of detecting fine-grained semantic difference, as shown in our experiments.

### 2.3 Multimodal Comprehension Evaluation of MLLMs

As a developed realm and valuable resource of high-quality data, VQA has been applied in the evaluation of MLLMs. MME [13] proposes a smart quantitative analysis of MLLMs with manually designed instruction-answer pairs that strictly limit responses to be *yes* or *no*. Therefore, all MLLMs are evaluated relatively fairly. However, although MME is insightful and effective, such detour avoids the problem of evaluating open-ended response directly. It ignores previous huge amount of VQA data and costs additional human annotation, limiting the scale of the dataset and making it tough for expanding. MM-vet [60] classifies VQA into the integration of multiple key abilities, and manually annotates corresponding VQA samples of their required abilities. Then, they use ChatGPT for evaluation (which we show to be less capable in evaluating the correctness of open-ended responses, refer to Section 5.3). The classification of VQA abilities is insightful and aids to the probing of specific abilities of MLLMs. Yet MM-vet requires high-quality annotation to identify the VQA abilities each question requires and are thus limited to a small amount too.

---

[2] As of the time of submission, Voyage-lite-02-Instruct achieves the best performance on the task of Semantic Textual Similarity (STS).

## 3 SEMANTIC EVALUATION OF VQA

With the rapid development of MLLMs, current metrics in VQA response evaluation are too stubborn to assess the rich generation and hinder evaluating MLLMs' performance with existing VQA datasets. Meanwhile, as mentioned in Section 2.2, current semantic evaluation models and tasks are inconsistent with the goal of flexible VQA response evaluation.

Therefore, this paper proposes the task of semantic evaluation of VQA, aiming at introducing flexible similarity-based soft evaluation with continuous scores into the assessment of VQA responses, contrary to inflexible metrics like Exact Match or VQA Score that require identical morphology of responses towards ground-truth labels. Such flexible evaluation enables to assess the rich generation from MLLMs and thus enables to use existing VQA datasets for probing MLLMs' multimodal comprehension ability.

### 3.1 Characteristics of VQA Evaluation

The proposed task of semantic VQA response evaluation shares significant difference with existing semantic evaluation tasks like STS and contains its own characteristics.

*Discrimination Granularity.* As mentioned in Section 1, traditional semantic evaluation tasks typically focus on the overall meaning in texts, rather than capturing the fine-grained detailed difference. However, the core of semantic VQA evaluation is comparing the response with annotated answer under the same question[3], where both texts share large overlap in meaning as the questions are same. Therefore, semantic VQA response evaluation demands fine-grained similarity discrimination.

*Text Length.* As VQA answers are generally much shorter than text in STS [4], n-gram based formulaic metrics like BLEU [44] will be more easily affected by the context in response. Model-based metrics are also vulnerable to such length shift, as their training data barely cover similar pattern.

*Distribution Shift.* The texts in STS datasets [2, 4, 5] come from general domains, like news and social media, while different VQA datasets comply to different sub-tasks, like knowledge [40] or reasoning [25]. Such distribution shift causes inconsistent evaluation on responses from different VQA datasets.

### 3.2 Three Key Properties in VQA Evaluation

To systematically evaluate the performance of a VQA evaluator, we propose three quantitative key properties, i.e., **Alignment**, **Consistency** and **Generalization**.

*Alignment.* Alignment assesses the overall performance of similarity scores predicted by evaluators with that of human annotation, in the metric of Spearman's Rank Correlation following similar setting in previous works[4, 5, 21, 32].

*Consistency.* A smart evaluator shall catch the key information in responses and ignore the noise text, e.g., the response of *elephants* shall be scored equally with *Theses are elephants* under the question of *What are the animals?*. Therefore, Consistency measures how close the different responses sharing the same meaning are scored.

*Generalization.* Considering various VQA datasets focus on various sub-tasks and come from various sources, Generalization depicts how well an evaluator is able to handle text from different domains. Refer to Section 3.4 for quantitative definitions.

### 3.3 A Dataset Assessing VQA Evaluators

To comprehensively compare and analyze the behaviour of different evaluators on VQA responses, taking the proposed three key properties into consideration, we propose a dataset Assessing VQA Evaluators (AVE). By collecting multiple MLLMs' responses on multiple datasets, the proposed dataset simulates a real scene of applying evaluators to evaluate the quality of various VQA responses. In order to compare the evaluators' scoring results with human judgement, we provide human annotation of the semantic correctness of responses towards ground-truth answers. The construction process of AVE is shown as follows:

*Response Collection.* First, we collect responses of five models, LLaVA [35], BLIP2 [30], mPLUG-Owl [58], OFA-large [53], Qwen-VL [8] on the validation set of four datasets, OKVQA[40], A-OKVQA[47], VQA v2[22] and GQA[25].

*Sampling Results.* Second, we sample in the responses while controlling the sampling amount of each dataset to be the same. In addition, samples that are answered correctly, i.e., the response is identical with the ground-truth answer, are excluded.

*Human Annotation.* Third, three annotators are asked to measure the semantic similarity[5] of each sampled response towards the ground-truth label and annotate an integral similarity score from 0 to 10, under certain rules (refer to Appendix for more details). Then the scores are averaged over the three annotators.

*Description Generation.* Fourth, in order to simulate MLLM responses with sentences instead of words or phrases, we select responses that are shorter than three words for augmentation. The augmentation contains two Parts. The first Part comes from using ChatGPT (refer to Appendix for prompts) to convert each pair of question and response into three descriptions and asking ChatGPT to select two descriptions that are closest to the origin question-answer pair as augmented responses. For example, the question of *What are the animals?* and the response of *elephants* are fed into ChatGPT, and generate descriptions like *The animals are elephants.* The second part comes from manually designed answer templates (refer to Appendix) to increase the diversity of descriptions rather than fully relying on ChatGPT. Now each sample contains three descriptions.

*Synonym Generation.* Fifth, we use WordNet [41] to locate a synonym for each answer. For cases where multiple synonyms exist,

---

[3]Considering the polysemy and ambiguity of words and phrases, the question text is indispensable for evaluating the semantic correctness.

[4]About 97.9% answers in VQA v2 [22], 97.7% in OKVQA [40], 99.9% in GQA [25] are shorter than three words. The average length of text in STS-12[4] is 12.5, which is much longer.

[5]We considered multiple aspects of measuring the correctness of a response towards the ground-truth answer, yet at last we come to the single aspect of semantic similarity for annotation. Refer to Appendix for more explanation.

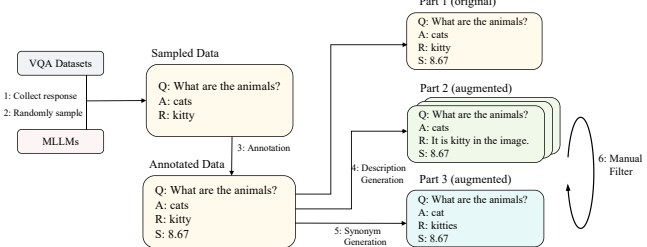

**Figure 2: The construction procedure of AVE. After randomly sampled from the outputs of models, each sample is manually annotated with a score and automatically augmented by generated descriptions and a variation on the answer word while remaining almost the same correctness as a VQA response. Different parts denote different augmentation methods.**

we choose the most common synonym by countering frequencies of words in Brown Corpus [20]. Meanwhile, we ask ChatGPT to introduce a shift in morphology to simulate cases that the outputs are merely different in tenses or singular or plural forms. The augmented answer is deemed to contain the same meaning with small disturbance on the style.

*Manual Filter.* At last, to ensure high-quality of the dataset, the three annotators also conduct manual filter (refer to Appendix for rules) to eliminate ambiguous samples, especially those generated from the fifth stage.

Generally speaking, the whole AVE dataset consists of three parts generated from above: Part 1 contains original answers and original responses. Part 2 contains original answers and the generated descriptions of responses. Part 3 contains tense-shifted answers and original responses. The whole procedure is depicted in Figure 2.

Meanwhile, the AVE dataset can also be clustered by the involved four datasets that each sample belongs to, OKVQA, A-OKVQA, VQA v2 and GQA, merging the Part 1 to Part 3 together and classifies by the sources of data only. Refer to Section 3.4 for how they are used.

The total sample amount of the final AVE dataset is 3,592, with each sample containing four types of augmentation results, as described above. The dataset is then split into a validation set and a test set with the ratio of 3:7. The distribution of annotated scores is shown in Appendix, which is relatively smooth. To evaluate the inter-annotator agreement, following previous works [11, 19, 23], we apply Krippendorff's alpha [28] and obtain a result of 0.713.

### 3.4 The Proposed Evaluation Indicators

In AVE dataset, a sample consists of a question $q_i$, a ground-truth label $a_i$, a source dataset label $d_i$, a response $r_i$ and a human-annotated score $s_i$. The task of VQA response evaluation can be defined as: given a question-answer pair, an evaluator $f(q_i, a_i, r_i)$ is expected to predict the annotated similarity score $s_i$ with the output $o_i$.

$$o_i = cos(f(q_i, r_i), f(q_i, a_i)) \quad (1)$$

$$score_f = Spearman(O, S) \quad (2)$$

where $cos$ stands for cosine similarity, $score_f$ is the performance score of the evaluator $f$, with $O$ and $S$ indicating the lists of all predicted scores and annotated scores respectively. Note that the metrics used for evaluating evaluators is Spearman's rank coefficient of correlation (Spearman).

The key properties of Alignment, Consistency and Generalization introduced in Section 3.2 are computed as follows:

*Alignment.* We use the average result of an evaluator on all parts of the proposed AVE dataset as alignment:

$$Alignment = \frac{1}{N_{Parts}} \sum_{i=1}^{N_{Parts}} score_{f_i} \quad (3)$$

where $N_{Parts}$ is the number of Parts in the AVE dataset and $N_{sets}$ is the number of involved VQA datasets in AVE, according to different type of division. The $score_{f_i}$ is the spearman score of evaluator $f$ on the $i$ th part of AVE.

*Consistency.* Consistency measures how close the responses of the same meaning with different morphology are evaluated. We regard the variance of the same sample among the three parts of AVE as consistency:

$$Consistency = \log \left(1/\left(\frac{1}{N_{samples}} \sum_{j=1}^{N_{samples}} var(o_{j_1}, o_{j_2}, o_{j_3})\right)\right) \quad (4)$$

where $N_{samples}$ is the amount of samples and var denotes calculating the variance. Then, $o_{j_1}, o_{j_2}, o_{j_3}$ are the predicted scores of the evaluator on Part 1, 2, 3 for the same sample $j$, respectively.

*Generalization.* Generalization measures the difference of performance on various datasets, and we define it as the variance of the performance on each involved VQA dataset:

$$\begin{aligned} Generalization = \log 1/\ var(align_{OKVQA}, \\ align_{A-OKVQA}, align_{VQAv2}, align_{GQA}) \end{aligned} \quad (5)$$

where $N_{sets}$ is the number of involved VQA datasets and $align_{dataset}$ is the mean Alignment score on the AVE data belonging to the corresponding VQA dataset.

## 4 SEMANTICALLY FLEXIBLE VQA EVALUATOR

With the three key properties of an ideal evaluator taken into consideration, we propose a novel evaluator based on meticulously designed pretraining tasks.

### 4.1 Pretraining Tasks

To guide the model to be sensitive to the key information between answer and response, this paper introduces several pretraining tasks to enhance the embedding. Data for augmentation come from a random sampling of VQA data in the training sets of OKVQA[40], A-OKVQA[47], TDIUC[27], VG-QA[29], GQA[25] and VQA v2 [22], which end to a total amount of 105, 311 samples. All augmented samples are mixed for training.

*NLI data.* In previous works [21, 32], models performs well with the natural language inference datasets SNLI [17] and MNLI [54], where each sample includes a premise, an entailment and a contradiction. In addition, these NLI datasets are all manually constructed, ensuring the high quality of their data, and the premise shares limited overlap with entailment compared with the sentence pairs in back-translation datasets. Therefore, to ensure the fundamental discriminating ability of models to capture overall meaning of sentences, we adopt NLI data and regard the premise-entailment pairs as postive pairs and premise-contradiction pairs as negative pairs.

*Candidate answers.* To make the best of available VQA datasets, for datasets with ten candidate answers, OKVQA, A-OKVQA and VQA v2, we consider candidate answers as correct answers as well. Then, for each sample, the most common candidate answer and a less common one are used to form a positive pair, with a random answer sampled from the answer space as negative.

*Synonym and Antonym.* In VQA response evaluation, semantically similar answers shall receive similar scores. We replace the answer with a synonym by WordNet [41], and if the antonym of an answer exists, we then pair up the answer and antonym as a negative pair, else we pair up the answer and a randomly sampled answer from the answer space as a negative pair. In addition, we use ChatGPT to produce synonyms as well, as ChatGPT is able to capture contextual information in the question and thus generates more accurate synonyms.

*Generated descriptions.* To simulate the output of MLLMs, for each sample, we provide ChatGPT with its question and answer to generate three descriptions with small disturbance of the same meaning. Then, we construct positive samples by pairing up the original answer and each generated description. For negative samples, we replace the answer in the generated description with a randomly sampled answer. The goal is to pull the embedding representation of a natural language description close to its simple form of a single answer, so that responses with different length but carrying similar meanings will receive similar scores. In addition, the negative pair is constructed by replacing the key answer word in the description, therefore guiding the model to be sensitive to the key words and to ignore the noise.

## 4.2 Model Framework

Following previous works [15, 16, 21, 32] on the STS task [1–5, 12], we use cosine similarity for distance calculation between embeddings[21, 32]. As shown in Figure 3, we adopt the simple contrastive learning framework [15] and contrastive learning with in-batch hard negatives [21].

The backbone encoder in this paper is RoBERTa [36]. In order to gain better generalization and comprehension ability, we apply the decoder-only LLM LLAMA2 [50] with the prompt[32] of *Summarize the text {text} in a single word:*. Then, the hidden states of the first generated new token is considered as the embedding vector.

The contrastive learning with in-batch hard negatives loss [21] is defined as follows:

$$loss_{ibn} = -\log \frac{e^{sim(h_i, h_i^+)/\tau}}{\sum_{j=1}^{N}(e^{sim(h_i, h_j^+)/\tau} + e^{sim(h_i, h_i^-)/\tau})} \quad (6)$$

where $h_i$ is the embedding representation of sample $i$, $h_i^+$ and $h_i^-$ respectively denote the representation of the positive sample and in-batch hard negative sample of sample $i$.

## 5 EXPERIMENTS

### 5.1 Implementation Details

Experiments in this paper is based on *transformer* package[55] on Pytorch. We use AdamW [37] optimizer, and the hyper-parameters of AdamW, betas, eps and weight-decay are set to 0.9, 0.999, 1e-8 and 0.01. We use a cosine scheduler and the batch size and peak learning rate for encoders are 128, 1e-5 for RoBERTa-base [36], VisualBERT [31] and LXMERT [49], 32, 6e-6 for RoBERTa-large and 8, 4e-6 for LLAMA2 [50].

### 5.2 Baselines

To assess to what extent existing models are competent for the VQA response evaluation, this paper collects four types of common methods for semantic similarity evaluation and refer to them as: formulaic, PLM, LLM, and API.

- Formulaic methods contain BLEU [44], ROUGE [33] and METEOR [10]. These methods base on n-grams for assessing the overlap. As VQA answers are usually short, we also report 2-gram results for BLEU and ROUGE.
- PLM refers to the Pretrained Language Models, which are generally small in sizes and typically in BERT-like encoder-only structures. SBERT (Sentence BERT) [45] embeds the text by BERT-like structures as the backbone and generate the text embedding vector, and apply cosine similarity to calculate the distance between vectors to decide the textual similarity. SIMCSE [21] provides both unsupervised and supervised methods, and this paper selects the supervised and better-performing one trained on NLI datasets for comparison. BGE [14] follows a multi-task learning scheme that collects and pretrains on multifarious datasets for better generalization. AnglE [32] aims to mitigate the gradient saturation issue encountered when using cosine distance by projecting vectors onto the complex plane and introducing an angular loss.
- LLM refers to large language models. This paper selects four of the well-performing LLMs, Baichuan2 [9], Qwen [7], LLAMA-2 [50] and Mistral [26].
- API refers to the remote usage of models by API online, including ChatGPT and Text-embedding-v3-large from OpenAI, and Voyage-lite-02-instruct from Voyage AI. Refer to Appendix for prompts. The latter two are embedding models that produce text embedding of given text, which are then used to calculate similarity score by cosine distance.

### 5.3 Main Experiments

Table 1 exhibits the performance comparison of various evaluators. The results are on the test set of AVE, with specific scores on each

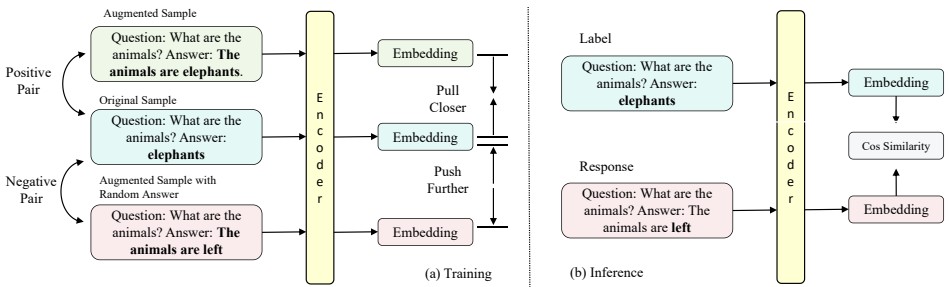

**Figure 3: Framework of contrastive learning in the proposed Semantically Flexible VQA Evaluator (SFVE). The original sample in augmented into two variations and form a positive pair and a negative pair. The example in the figure shows the procedure of the pretraining task *Generated descriptions*. In the positive pair, the semantics of the sentence is considered same as the original, while in the negative pair, as the answer word is replaced with a random answer, the sentence contains unmatched meaning with the original.**

| Types | Methods | Alignment ↑ | | | | Consistency ↑ | Generalization ↑ | STS Avg. ↑ |
|---|---|---|---|---|---|---|---|---|
| | | Part 1 | Part 2 | Part 3 | Avg. | | | |
| Formulaic | BLEU-2 | -1.7 | -2.6 | 3.2 | -0.4 | 4.35 | 11.30 | 50.1 |
| | BLEU-4 | -2.9 | -3.5 | 2.0 | -1.5 | 4.23 | 11.25 | 47.6 |
| | ROUGE-2 | -2.8 | -4.5 | 1.2 | -2.0 | 5.79 | 11.82 | 53.9 |
| | ROUGE-L | 4.9 | -0.1 | 3.3 | 2.7 | 6.73 | 10.78 | 48.3 |
| | METEOR | 12.4 | 4.4 | 15.3 | 10.7 | 7.25 | 9.34 | 53.4 |
| PLM | RoBERTa-large (w/o CL) | 11.9 | 0.7 | 23.4 | 12.0 | 7.91 | 10.36 | 27.9 |
| | SBERT | 47.7 | 44.3 | 40.6 | 44.2 | 8.79 | 8.55 | 76.8 |
| | SIMCSE | 44.9 | 44.7 | 41.7 | 43.7 | 9.37 | 8.30 | 83.8 |
| | BGE | 42.3 | 36.5 | 41.0 | 39.9 | 8.93 | 8.25 | 84.9 |
| | AnglE | 43.4 | 38.2 | 40.2 | 40.6 | 9.01 | 7.78 | 86.4 |
| LLM | Baichuan2-7b | 28.1 | 27.8 | 31.8 | 29.2 | 5.30 | 8.86 | 64.6 |
| | Qwen-7b | 25.9 | 26.3 | 24.2 | 25.5 | 9.07 | 10.10 | 68.3 |
| | LLaMA2 | 32.7 | 27.9 | 34.6 | 31.7 | 7.45 | 8.53 | 61.9 |
| | Mistral-7b | 16.8 | 14.5 | 20.7 | 17.3 | 4.61 | 8.49 | 72.1 |
| API | ChatGPT | 21.2 | 15.2 | 24.6 | 20.3 | 5.21 | 8.35 | 73.7 |
| | Text-embedding-v3-large | 32.5 | 28.6 | 36.3 | 32.5 | 9.40 | 8.00 | 82.3 |
| | Voyage-lite-02-instruct | 29.1 | 28.9 | 29.3 | 29.1 | 11.81 | 6.78 | 86.3 |
| SFVE (ours) | SFVE-base | 58.4 | 57.1 | 53.7 | 56.4 | 9.12 | 8.34 | 81.2 |
| | SFVE-large | 58.1 | 57.5 | 56.0 | 57.2 | 9.53 | 8.67 | 82.0 |
| | SFVE-LLAMA2-7b | 60.2 | 57.0 | 57.2 | 58.1 | 9.46 | 8.87 | 77.9 |

**Table 1: The comparison of performance on our proposed AVE dataset. The STS Avg. denotes the average scores over STS 2012 to STS 2016 [1–5], SICK-R [39] STS-B [12], providing a reference of methods' general discriminating ability. RoBERTa-large [36] (w/o CL) refers to the original pretrained checkpoint without contrastive learning. SFVE-base, SFVE-large and SFVE-LLAMA2-7b are RoBERTa-base, RoBERTa-large and LLAMA2-7b trained by contrastive learning on our proposed pretraining tasks. The specific model checkpoints in experiments are as follows: SBERT[45]: SRoBERTa-NLI-large, BGE[14]:BAAI-bge-large-en, SIMCSE[21]: RoBERTa-NLI-large, AnglE [32]: RoBERTa-large.**

part of the dataset, as described in Section 3.3. To promote a comprehensive assessment of existing methods, this paper compares the performance with four common types of methods for semantic evaluation, as introduced in Section 5.2. The last row of Types contains our results from training with the proposed pretraining tasks on the corresponding model. Then, the column of Alignment contains the separate results on each of the three parts in AVE datasets and their average.

*5.3.1 Performance of Formulaic Methods.* Formulaic methods, i.e., BLEU, ROUGE and METEOR perform poorly in Alignment scores, and some of them drop below 0, indicating adverse scores to the human annotation. Such phenomenon is expected, as the n-gram matching strategy of BLEU and ROUGE is unable to handle the synonyms or variations in tenses and singular or plural forms. For

METEOR, however, it applies port stem [10] and synonym matching to preprocess the 1-gram in both the candidate and reference, restoring words to stems and thus performs better.

In addition, it is interesting to notice that although the Alignment of BLEU and ROUGE are much lower that that of METEOR, their Generalization scores are much higher. There are two reasons to this anomaly. First, BLEU and ROUGE fail to handle the task well and their prediction can be considered random, thus the sources of data do not affect the results, just like RoBERTa-large (w/o CL). Second, these n-gram based evaluators do not involve semantics, therefore the sources of data that causes word distribution shift matter less.

*5.3.2 Performance of PLMs.* The BERT-like models pretrained for textual similarity prediction, i.e., SBERT, SIMCSE, BGE, AnglE (the

| Settings | Alignment ↑ | | | | Consistency ↑ | Generalization ↑ |
|---|---|---|---|---|---|---|
| | Part 1 | Part 2 | Part 3 | Avg. | | |
| All tasks | 58.4 | 57.1 | 53.7 | 56.4 | 9.12 | 8.34 |
| w/o *NLI data* | 53.3 | 52.3 | 50.9 | 52.2 | 8.71 | 8.50 |
| w/o *Candidate answers* | 57.3 | 56.5 | 53.0 | 55.6 | 9.33 | 8.54 |
| w/o *Synonym and Antonym* | 42.1 | 40.3 | 38.8 | 40.4 | 9.19 | 8.41 |
| w/o *generated descriptions* | 56.9 | 47.0 | 52.3 | 52.1 | 8.07 | 8.93 |
| w/o *All tasks* | 12.5 | 3.1 | 20.0 | 11.8 | 10.58 | 9.90 |
| *NLI data* only | 44.3 | 42.0 | 33.1 | 39.8 | 9.46 | 10.00 |
| *Candidate answers* only | 37.4 | 29.8 | 39.6 | 35.6 | 8.10 | 8.11 |
| *Synonym and Antonym* only | 53.8 | 43.7 | 50.6 | 49.4 | 7.89 | 9.08 |
| *Generated descriptions* only | 42.8 | 49.1 | 42.1 | 44.6 | 8.17 | 7.67 |

Table 2: Ablation experiments of designed pretraining tasks on RoBERTa-base. The row of All tasks represents the best performance of RoBERTa-base with all pretraining tasks, and the row of w/o All tasks contains results from testing on the RoBERTa-base checkpoint without further training. w/o represents without the corresponding pretraining task, contrary to the setting in lower part of the table where the model is trained only on a single task each time.

latter four models), show much better performance than RoBERTa-large (w/o CL) and formulaic methods, indicating the basic textual similarity tasks are helpful to the VQA response evaluation task, but they fail to align well with human judgement, compared to the SFVE results under the same structure of BERT-large. In addition, the performance on Part 1 and 3 of the latter four PLM models are similar, and the major gap lies in the capability of processing long responses.

*5.3.3 Performance of LLMs.* For LLMs (refer to Appendix for the detailed prompt) including ChatGPT, they fail to gain satisfactory results on AVE. Naturally, LLM performs better than RoBERTa w/o CL, and Generalization scores are slightly higher than PLMs, which we attribute to the better generalization ability of LLMs. Although LLMs obtain acceptable results on STS tasks, just like the formulaic methods, they encounter significant performance drop on the VQA response evaluation. Such phenomenon verifies the significant difference between the evaluation of STS and VQA responses and the necessity in the task of VQA evaluation.

The performance of embedding models (the latter two models) on STS is higher than LLMs but the VQA response evaluation performance is still low. The reason to their incompetence on AVE, we speculate, is that these embedding models focus more on retrieving and capturing the general meaning of given texts than discovering the fine-grained difference between given pair of texts. In addition, such focus of capturing the general meaning of texts has also empowered them with the ability to ignore noise in morphology and text length, thus gaining high scores of Consistency despite the low Alignment scores.

*5.3.4 Performance of SFVE.* The section of SFVE (ours) in the table presents our results on AVE. The pretraining tasks effectively improve the Alignment scores of all three models and bring moderately better Consistency and Generalization performance. From the prospective of model sizes, the 125M Roberta-base demonstrates similar capability with the 355M Roberta-large with merely a gap of 0.8%. The same applies for the 7b LLAMA2, which surpasses RoBERTa-large by 0.9%. Giant increase in model sizes brings minor improvement in scores. We believe the reason is that the similarity measure, either in STS or AVE, is relatively simple for models to comprehend and implement, where a simple structure with limited

parameters is able to achieve excellent performance with proper training.

Therefore, during the training of generative VQA models, considering the significantly larger computation cost in LLAMA-7b than RoBERTa, we recommend utilizing RoBERTa base or large for a rough validation of model performance each certain steps or epoch, and use LLAMA for more accurate evaluation near the best steps or epochs.

## 5.4 Ablation Experiments

To analyze the influence of each pretraining task, Table 2 provides ablation results by removing a pretraining task each time and by training on a single task alone. From the table it is clear that all pretraining tasks contribute to the final performance more or less.

The most important task is *Synonym and Antonym*, which causes a drop of 16.0% in Alignment scores on average and damages Consistency as well. In addition, when trained only on such data, the model performs the best. We believe the importance of training on *Synonym and Antonym* task lies in aligning the representation of synonyms and increasing the difference towards antonyms and other answers.

The second influencing pretraining task is *Generated descriptions*, without which the model can not directly learn to align the representation between semantically similar texts with different length. Yet the removal of it does not substantially damage the results on other parts than Part 2, which consists of long responses.

Meanwhile, the removal of *NLI data* matters almost the same as *Generated descriptions*. As mentioned before, NLI data focuses more on the coarse-grained meaning between text pairs while AVE requires a finer semantic discrimination. However, for a model that barely handles the task (shown in the row of *w/o All tasks*), we believe the easier data in NLI aid to fertilizing the basic capability in semantic evaluation. Yet the NLI data alone is insufficient, as shown in the row of *NLI data only*.

## 5.5 Practical Application

To demonstrate the practical values of our proposed evaluator in flexible VQA response evaluation, we collect responses of multiple MLLMs and compare the results with different evaluators by overall scores and case study.

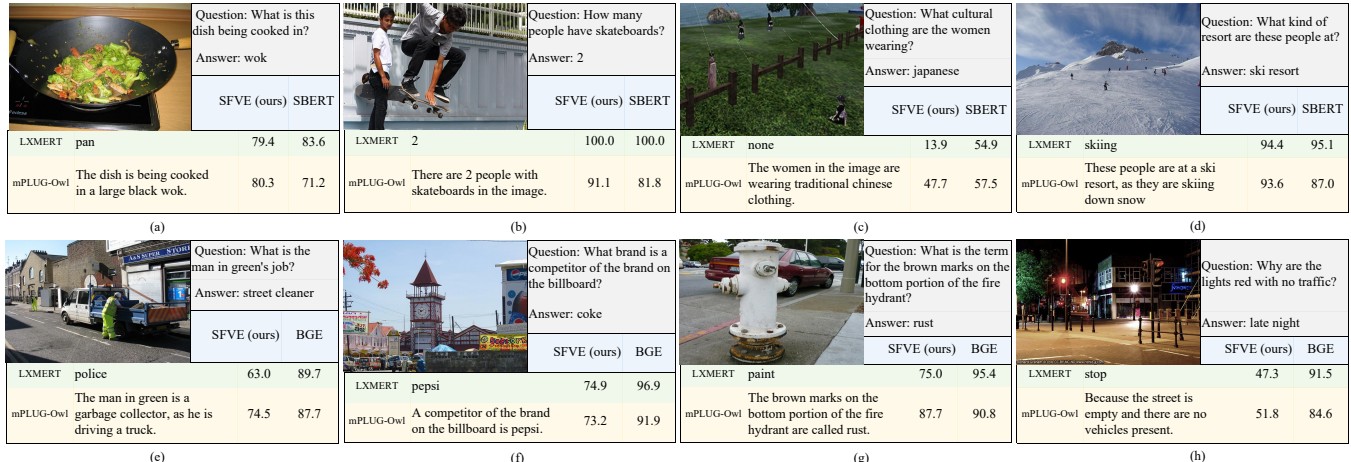

**Figure 4: Cases for analysis. The samples come from the open-ended part of A-OKVQA [47] validation set. The first row comes from results of SFVE and SBERT, and the second comes from SFVE and BGE.**

| Model | Evaluation Metric | | | |
|---|---|---|---|---|
| | VQA Score | BGE | SBERT | SFVE (ours) |
| LXMERT [†] [49] | 19.5 | 83.3 | 75.6 | 43.6 |
| LXMERT | 37.3 | 94.9 | 83.6 | 67.9 |
| VisualBERT [31] | 37.6 | 94.8 | 83.4 | 66.3 |
| LLaVA-7b [35] | 3.6 | 89.1 | 82.5 | 72.3 |
| BLIP2-opt-2.7b [30] | 15.5 | 94.1 | 83.1 | 70.2 |
| InstructBLIP-Vicuna [18] | 21.4 | 94.8 | 86.4 | 74.3 |
| mPLUG-Owl [58] | 0.0 | 91.0 | 82.7 | 69.1 |
| OFA-large [53] | 39.5 | 95.3 | 86.5 | 78.0 |
| Qwen-VL-chat [8] | 54.9 | 96.1 | 89.7 | 83.5 |

**Table 3: Practical application of utilizing our proposed evaluator for assessing the responses from MLLMs. The VQA dataset for response generation is the open-ended validation set of A-OKVQA[47]. Models in the upper part of the table are smaller than 0.5B. VisualBERT and LXMERT are fine-tuned on VQA v2[22]. LXMERT [†] means the LXMERT that is not sufficiently trained, which ends training at the half of the first epoch to provide comparison. SFVE (ours) uses the RoBERTa-large evaluator trained with our proposed pretraining tasks. Refer to Appendix for the calculation of VQA Score. Note that the scores are for comparison within the an evaluator itself, and it is meaningless to compare scores across evaluators, as evaluators are not aligned.**

As shown in Table 3, VQA score is clearly incompetent for assessing assorted responses from MLLMs. Since all responses from mPLUG-Owl are sentences, VQA Score even comes to 0. In the comparison of LXMERT and mPLUG-Owl, both BGE and SBERT indicate LXMERT generates better responses than mPLUG-Owl. However, taking the case study in Figure 4 into consideration, we verify that existing well-performing methods, BGE and SBERT, fail to perform consistent evaluation and bias towards short responses while penalizing longer ones. For example, in (a) of Figure 4, LXMERT response *pan* receives a much higher score than the mPLUG-Owl response which is a descriptive sentence containing

the correct answer. In (c), the descriptive text and the single word response receive similar scores under our SFVE, but SBERT considers the short answer of LXMERT is much better than the descriptive sentence of mPLUG-Owl. Similar phenomena exist in BGE as well. As in (e), mPLUG-Owl response describing *garbage collector* is a much better than LXMERT output *police*, yet the latter receives even higher scores.

In addition, not only does the length impede a fair evaluation, but the incompetence in fine-grained semantics discrimination also causes absurd results. Like in (c) and (g), where LXMERT answers are less correct but they receive competitive or even higher scores than reasonable responses. Such error, we speculate, is caused by focusing more on the overall meaning of text, as the questions are the same within a pair.

Due to the phenomena above, it is clear the superficial superiority of LXMERT over mPLUG-Owl is merely a mistake by incompetent VQA evaluators, which also demonstrates the importance of fairness and consistency in VQA evaluation. We consider the proposed pretraining tasks and SFVE effective, not only on our proposed AVE dataset, but also in practical application where previous methods fail to perform fair and insightful evaluation.

## 6 CONCLUSION

This paper proposes a practical task of utilizing semantic correctness to evaluate unconstrained open-ended VQA responses, facilitating the assessment of MLLMs' multimodal comprehension abilities by VQA data. We propose three key properties for assessing VQA evaluators, i.e., Alignment, Consistency and Generalization. In addition, this paper proposes a new dataset assessing VQA evaluators (AVE) to comprehensively analyze multiple aspects of evaluators. Based on contrastive learning with meticulously designed pretraining tasks, this paper provides a Semantically Flexible VQA Evaluator (SFVE) that performs significantly better than existing evaluators on VQA evaluation and the training scheme generalizes to both the encoder-only and decoder-only models.

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
