# OpenReview forum: "Towards Flexible Evaluation for Generative Visual Question Answering"
_acmmm.org/ACMMM/2024/Conference — MM2024 Oral_

### Official Review · Reviewer_BQYv · 2024-05-12

**Rating:** 5
**Confidence:** 3

**Summary:**

The paper addresses the important task of the evaluation of generative models for VQA. The technical contribution is twofold: the introduction of the AVE dataset for benchmarking VQA evaluators, and a new family of evaluators trained with a carefully derived pre-training strategy (SFVE). Both the dataset and the models leverage data augmentation schemes, empirically validated, derived from the three principles of Alignment, Consistency, and Generalization.

**Strengths:**

- the paper is generally well-written, the motivations are clear, and the task is of high relevance, given that generative VQA is garnering importance, especially thanks to the emergence of MLLMs
- the experimental results are strong, and the ablative analysis validates the key design choices of the paper

**Limitations:**

- clarification on description generation. L330 states that only answers shorter than three words are selected for augmentation (description generation in this case). But from L340 and L390 (*...each sample containing four types of augmentation...*, I imagine those being 1) description generation from ChatGPT and 2) templated answers, 3) synonym generation from WordNet and 4) ChatGPT) I understand that each sample is augmented with all four strategies. Can you clarify this point? Are answers longer than three words augmented with description generation as well? In case of no, the precise amount of augmented samples may be disclosed for completeness.
- L445: what does **N_sets** refer to?
- the four datasets used to build AVE are popular and established. However, more recent VQA datasets, with a particular focus on knowledge-intensive questions, would benefit the proposed benchmark in terms of generalization [1, 2].
- for completeness, Table 1 should include the comparison between SVFE-base and "RoBERTa-base (w/o CL)"
- other models specifically designed for Q-A evaluation may be considered in the comparison, e.g. [3]

[1] Chen, Yang, et al. "Can Pre-trained Vision and Language Models Answer Visual Information-Seeking Questions?." EMNLP 2023

[2] Mensink, Thomas, et al. "Encyclopedic VQA: Visual questions about detailed properties of fine-grained categories." ICCV 2023

[3] Bulian, Jannis, et al. "Tomayto, Tomahto. Beyond Token-level Answer Equivalence for Question Answering Evaluation." EMNLP 2022

**Suitability:**

3

---

### Official Review · Reviewer_pbVk · 2024-05-14

**Rating:** 3
**Confidence:** 4

**Summary:**

The paper proposed the use of semantics-based evaluators for evaluating MLLM responses. In particular, the evaluation relied on three key properties, *i.e.*, Alignment, Consistency and Generalization. The authors proposed a corresponding dataset (AVE) to facilitate the analysis.

**Strengths:**

* The paper is well-written, devoid of grammatical errors, and exhibits smooth readability.
* The task is perfectly aligned with the current demands within the domain of MLLM.
* I greatly appreciate the proposition of a novel benchmark for evaluating MLLM.
* The experiments conducted are clear, and the analysis of these experiments is articulated.

**Limitations:**

* The structure of the paper is sometimes confusing. A lot of space has been dedicated to defining the new benchmark, neglecting the evaluation model. I would appreciate more details regarding this.
* Despite its effectiveness to provide a fair evaluation of the MLLM, it is evident that the overall contribution has deficiencies:
1. The proposed dataset exhibits a limitation in its scale. From my standpoint, especially regarding MLLMs, the direction for devising new benchmarks should lean towards large-scale datasets, precisely due to the inherent variance contained within these models of significant dimensions.
2. Lack of comparisons with Vision and Language metrics relied on embedding spaces, *i.e.* CLIP-S and variants.
3. The model mainly works with language only. But, not including visual features could make evaluations less accurate. When we describe images with text, we often miss fine-grained details. I believe this is a crucial point to think about.

**Suitability:**

3

---

### Official Review · Reviewer_mXzt · 2024-05-24

**Rating:** 6
**Confidence:** 4

**Summary:**

This paper proposes using semantic similarity-based evaluation metrics to assess open-ended responses in Visual Question Answering (VQA) datasets, in order to better evaluate the multimodal comprehension abilities of large language models. It defines three key properties for VQA evaluators - Alignment, Consistency and Generalization. The paper introduces a new dataset called AVE to comprehensively analyze different evaluators on these properties. It also proposes a new evaluator called SFVE that uses carefully designed pretraining tasks and contrastive learning to outperform existing semantic similarity evaluators on the VQA response evaluation task.

**Strengths:**

1. Addresses the limitations of current metrics like Exact Match and VQA Score in evaluating the rich open-ended responses generated by multimodal large language models. Enables leveraging existing VQA datasets.
2. Defines three important quantitative properties and provides a high-quality human annotated AVE dataset for systematically comparing and analyzing VQA evaluators.
3. The proposed SFVE evaluator significantly outperforms existing semantic similarity methods including ChatGPT and the SOTA Voyage-lite-02-Instruct. The pretraining scheme generalizes to both encoder-only and decoder-only models.
4. Demonstrates the practical value of the proposed flexible semantic evaluator in assessing real VQA model responses more fairly compared to metrics like VQA Score.

**Limitations:**

1. While the paper motivates the need for semantics-based evaluation well, it does not provide an intuitive sense of how the proposed metrics like Alignment, Consistency and Generalization scores map to downstream VQA model performance and capabilities.
2. The ablations, while useful, do not provide insights into the relative importance of the different pretraining tasks and augmentations for SFVE's performance gains. Further analysis could help refine the approach.
3. The paper is missing commentary on the computational overheads and feasibility of the proposed evaluation approach at larger scales, especially when leveraging giant LLMs.
4. Miss some related work.[1][2][3]

[1] Ying, Kaining, et al. "MMT-Bench: A Comprehensive Multimodal Benchmark for Evaluating Large Vision-Language Models Towards Multitask AGI." arXiv preprint arXiv:2404.16006 (2024).

[2] Shao, Wenqi, et al. "Tiny lvlm-ehub: Early multimodal experiments with bard." arXiv preprint arXiv:2308.03729 (2023).

[3] Liu, Shuo, et al. "ConvBench: A Multi-Turn Conversation Evaluation Benchmark with Hierarchical Capability for Large Vision-Language Models." arXiv preprint arXiv:2403.20194 (2024).

**Suitability:**

3

---

### Official Review · Reviewer_FoTc · 2024-05-25

**Rating:** 4
**Confidence:** 4

**Summary:**

The authors of the paper "Towards Flexible Evaluation for Generative Visual Question Answering" aim to address the limitations of evaluation methods for Visual Question Answering (VQA) models. Specifically, current metrics that rely on sentence syntax and exact matching do not align well with modern generative models. This leads to inaccurate evaluations that disadvantage rich and nuanced responses. To overcome this, the authors propose a manually annotated dataset based on three key principles: alignment, consistency, and generalization.

**Strengths:**

- This addresses an issue already present in image captioning: how to evaluate generative models. Specifically, it tackles how to assess generated responses that are semantically close to the ground truth (GT) but differ syntactically.
- The motivations proposed by the authors of the paper are increasingly relevant and contribute to the literature on evaluating responses generated by Multimodal LLMs. Creating a dataset in this direction can help guide future work.

**Limitations:**

- During the annotation process, only three annotators were used. This raises concerns about alignment with human judgment. A user study with a larger number of participants on a subset of the dataset could address this potential issue.
- An analysis is missing to understand the distribution and delta between the responses of the three annotators, which would provide insight into whether such a small number of annotators is sufficient for generalization and alignment.
- In line 343, when introducing data augmentation for captions, it would be interesting to have some quantitative or qualitative data to understand its effect.
- In constructing SFVE (ours), the visual part, with its features, is not considered.
- As highlighted in line 750, the models considered in Table 1 have very different parameter counts. Therefore, BERT models could be compared with a similarly sized LLM decoder-only model (3B, 125M) [1].
- The tested LLMs are either encoder-only or decoder-only. To have a complete spectrum, encoder-decoder models are missing. To address this, the performance of a model like T5 [2] could be analyzed.
- As noted in line 102, forcing a Multimodal LLM to generate only one word can be limiting. It would be useful and would enhance the contribution of the paper if the differences in performance were shown by demonstrating generative capabilities with and without constraints.
- The proposed dataset is small due to the extensive human annotation required for its development.

[1] Zhang et al. “Opt: Open pre-trained transformer language models”, 2022.

[2] Raffel et al. “Exploring the limits of transfer learning with a unified text-to-text transformer”, 2020.

**Suitability:**

2

---

### Meta-Review · Area_Chair_sfJZ · 2024-06-27

**Recommendation:** Accept (Oral)
**Confidence:** 5

**Metareview:**

The paper addresses the evaluation of generative models for Visual Question Answering (VQA) by introducing the AVE dataset for benchmarking VQA evaluators and a new family of evaluators trained with a carefully derived pre-training strategy called SFVE. After author rebuttal, this paper received overall positive ratings . All the reviewers consider the paper is well written and easy to follow. Strong experimental results and ablative analysis validate the key design choices of the paper. We recommend acceptance.